# Propofol-based intravenous anesthesia is associated with better survival than desflurane anesthesia in pancreatic cancer surgery

Hou-Chuan Lai[1], Meei-Shyuan Lee[2], Yin-Tzu Liu[3], Kuen-Tze Lin[4], Kuo-Chuan Hung[5], Jen-Yin Chen[5,6], Zhi-Fu Wu[5]*

1 Department of Anesthesiology, Tri-Service General Hospital and National Defense Medical Center, Taipei, Taiwan, Republic of China, 2 School of Public Health, National Defense Medical Center, Taipei, Taiwan, Republic of China, 3 Division of Anesthesiology, Wanfang Hospital, Taiwan, Republic of China, 4 Department of of Radiation Oncology, Tri-Service General Hospital and National Defense Medical Center, Taipei, Taiwan, Republic of China, 5 Department of Anesthesiology, Chi Mei Medical Center, Tainan City, Taiwan, Republic of China, 6 Department of the Senior Citizen Service Management, Chia Nan University of Pharmacy and Science, Tainan City, Taiwan, Republic of China

* aneswu@gmail.com

**Data Availability Statement:** All relevant data are within the manuscript and its Supporting Information files.

## Abstract

### Background

Previous researches have shown that anesthetic techniques can influence the patient outcomes of cancer surgery. Here, we studied the relationship between type of anesthetic and patient outcomes following elective, open pancreatic cancer surgery.

### Methods

This was a retrospective cohort study of patients who received elective, open pancreatic cancer surgery between January 2005 and July 2018. Patients were grouped according to the anesthesia they received, namely desflurane or propofol. A Kaplan–Meier analysis was conducted, and survival curves were presented from the date of surgery to death. Univariable and multivariable Cox regression models were used to compare hazard ratios for death after propensity matching. Subgroup analyses were performed for all-cause mortality, cancer-specific mortality, and disease progression.

### Results

A total of 68 patients (56 deaths, 82.0%) under desflurane anesthesia, and 72 patients (43 deaths, 60.0%) under propofol anesthesia were included. Fifty-eight patients remained in each group after propensity matching. The propofol anesthesia was associated with improved survival (hazard ratio, 0.65; 95% confidence interval, 0.42–0.99; $P = 0.047$) in the matched analysis. Subgroup analyses showed significantly better cancer-specific survival (hazard ratio, 0.63; 95% confidence interval, 0.40–0.97; $P = 0.037$) in the propofol group. Additionally, patients under propofol had less postoperative recurrence, but not fewer

**Funding:** The author(s) received no specific funding for this work.

**Competing interests:** The authors have declared that no competing interests exist.

**Abbreviations: ASA**, American Society of Anesthesiology; **CCI**, Charlson comorbidity index; **Ce**, effect-site concentration; **CI**, confidence interval; **ETCO2**, end-tidal carbon dioxide; **HCC**, hepatocellular carcinoma; **HIF**, hypoxia-inducible factor; **HR**, hazard ratio; **IRB**, institutional review board; **METs**, metabolic equivalents; **NSAIDs**, non-steroidal anti-inflammatory agents; **PS**, propensity score; **SD**, standard deviation; **TCI**, target controlled infusion; **TNM**, tumor–node–metastasis; **TSGH**, Tri-Service General Hospital; **VAs**, volatile anesthetics.

postoperative metastases formation, than those under desflurane (hazard ratio, 0.55; 95% confidence interval, 0.34–0.90; $P$ = 0.028) in the matched analysis.

## Conclusions

In a limited sample size, we observed that propofol anesthesia was associated with improved survival in open pancreatic cancer surgery compared with desflurane anesthesia. Further investigations are needed to inspect the influences of propofol anesthesia on patient outcomes of pancreatic cancer surgery.

## Introduction

Pancreatic cancer is one of the most fatal cancers in humans, and it may be the second leading cause of cancer death by the year 2030 [1]. In Taiwan, the incidence of pancreatic cancer is increasing as it is in other Western countries, with an incidence of 10–11/100,000 persons [1]. The most common histological type of pancreatic cancer is adenocarcinoma [1]. Pancreatic cancer carries the poor prognosis with a median survival of 6 months, and a 5-year survival rate is only around 5% [1]. Surgical resection plays an important role on the treatment for many cancers, including pancreatic cancer [2]. However, surgical intervention may result in neuroendocrine and metabolic changes, which may lead to impairment of cell-mediated immunity and activate the implantation of circulating tumor cells [3]. This potential combination of impaired immune responses and cancer cell seeding enhances the susceptibility of patients undergoing cancer surgery to the development of postoperative metastasis, and is associated with poor survival. The potential role of anesthetic techniques in the process of postoperative recurrence or metastasis formation has attracted attention [3].

Data from human cancer cell lines and animal researches showed that different anesthetics might affect the immune system in different paths [4–9]. Researches had shown that volatile anesthetics (VAs) were pro-inflammatory and might affect immune processes, which might increase the incidence of postoperative metastasis [8–12]. However, propofol seemed to reduce tumor growth and to decrease the risk of metastasis in humans and mice [6,11–14].

Call et al. [2] have reported that perioperative dexamethasone administration was associated with improved survival in patients undergoing pancreatic cancer surgery. Moreover, Soliz et al. [15] showed that propofol-based anesthesia was associated with lower postoperative complications compared with desflurane-based anesthesia in pancreatic cancer surgery. Until now, very few studies have compared the effects of the use of desflurane versus propofol anesthesia on patient outcomes after pancreatic cancer surgery. We hypothesized that patients under desflurane anesthesia might have subsequent poor outcomes than patients under propofol anesthesia as our previous colon cancer and hepatocellular carcinoma (HCC) studies [16,17]. Thus, we performed a retrospective cohort study to inspect whether the choice of the anesthetic, desflurane versus propofol is associated with patient survival, postoperative recurrence, and postoperative metastases formation following pancreatic cancer surgery.

## Methods

### Study design and setting

This retrospective cohort study was performed at the Tri-Service General Hospital (TSGH), Taipei, Taiwan, Republic of China.

## Participants and data sources

The ethics committee of the TSGH approved this retrospective study and waived the need for informed consent (TSGHIRB No: 2-106-05-101 and TSGHIRB No: 2-108-05-009). The information was retrieved from the electronic database and medical records of TSGH. From January 2005 to July 2018, 140 consecutive cases with an American Society of Anesthesiologists (ASA) score of II–III who had received elective, open resection for tumor-node-metastasis (TNM) of stage I–IV pancreatic adenocarcinoma under propofol anesthesia (n = 72) or desflurane anesthesia (n = 68) were eligible for analysis. The anesthetic technique was decided by the anesthesiologist's personal preference. The exclusion criteria were propofol anesthesia combined with VAs or regional analgesia, incomplete data, age < 20 years, and laparoscopic surgery. And then, 7 cases were excluded (Fig 1).

No medication was used before the anesthesia induction. Standard monitoring, including electrocardiography (lead II), noninvasive blood pressure testing, pulse oximetry, end-tidal carbon dioxide (ETCO2) measurement, the central venous catheter insertion, and direct radial arterial blood pressure monitoring, was performed in each case. Anesthesia was induced using fentanyl, propofol, and cisatracurium or rocuronium in all cases.

Anesthesia was maintained with target-controlled infusion (TCI) (Fresenius Orchestra Primea; Fresenius Kabi AG, Bad Homburg, Germany) using propofol at an effect-site concentration (Ce) of 3–4 μg/mL in the propofol group. Patients with propofol anesthesia received FiO2 of 100% oxygen at a flow rate of 300 mL/min. The desflurane vaporizer was set between 4% and

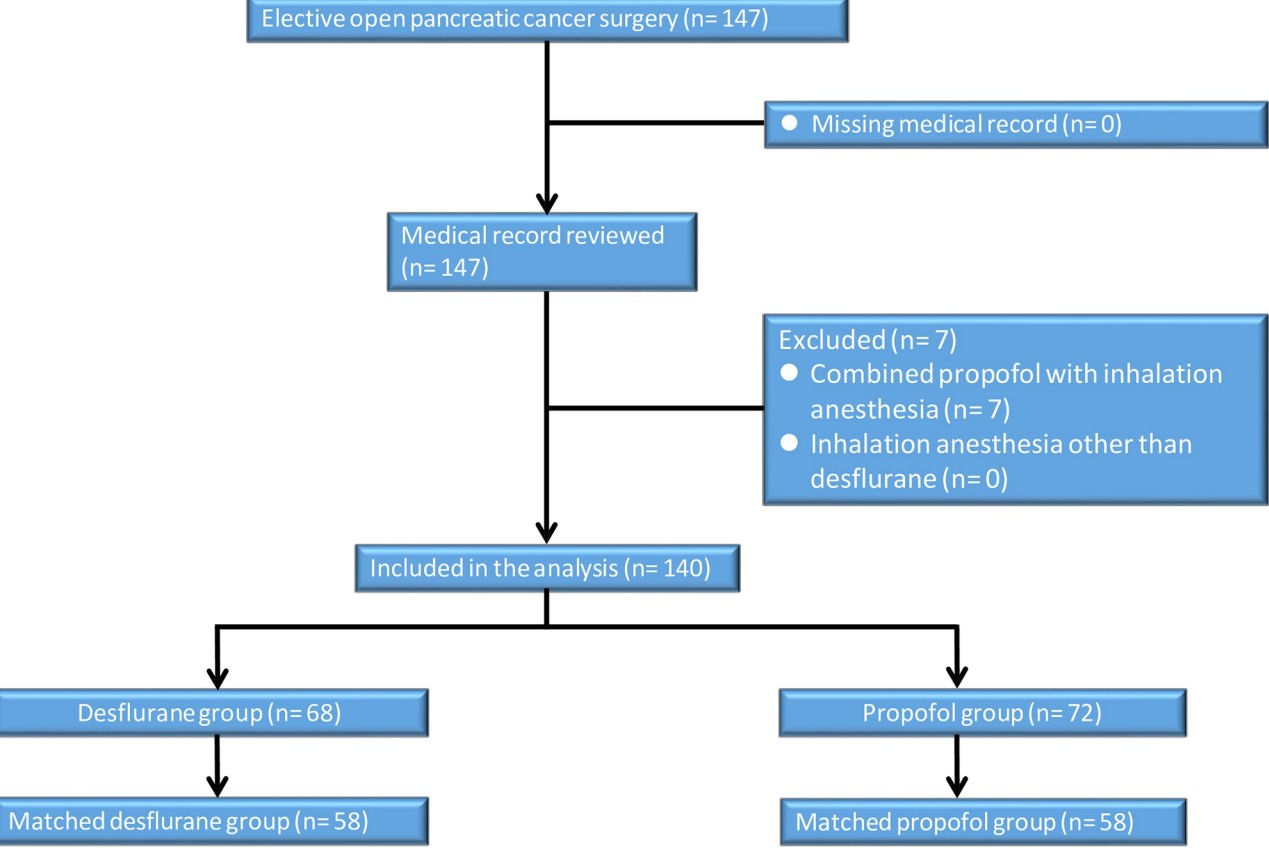

**Fig 1. Flow diagram detailing the selection of patients included in the retrospective analysis.** 7 patients were excluded due to combined propofol anesthesia with inhalation anesthesia or regional analgesia, incomplete data, age <20 years, and laparoscopic surgery.

10% in 100% oxygen at a flow of 0.3 L/min in a closed breathing system in the desflurane group. Repetitive bolus injections of fentanyl and cisatracurium were used as needed during surgery [16–18]. Desflurane or maintenance of the Ce with TCI using propofol was adjusted downward and upward by 0.5–2% or 0.2–0.5 μg/mL, respectively, when needed based on the hemodynamics. The level of ETCO2 was kept at 35–45 mmHg by adjusting the ventilation rate with a maximum airway pressure < 30 cm $H_2O$. After surgery, cases were transferred to the postanesthesia or intensive care unit and evaluated by the anesthesiologist in charge [16–18].

## Variables

We retrospectively gathered the following patient data: anesthetic technique; time since the earliest included patient, which served as a surrogate of the calendar year; calendar period; sex; age at the time of surgery; and preoperative serum CA19-9 values. For preoperative CA19-9 levels, patients were grouped according to whether their CA19-9 levels were > 37 or ≤ 37 U/mL, because a CA19-9 level > 37 U/mL is associated with poor survival in pancreatic adenocarcinoma [19]. We used the Charlson Comorbidity Index (CCI) to predict the 10-year survival in patients with multiple comorbidities. The preoperative functional capacity was assessed in metabolic equivalents (METs). Because the cardiac and long-term risks increase in patients with a functional capacity of < 4 METs during most normal daily activities [20], and patients were grouped according to whether the value was ≥ 4 METs or < 4 METs. We also used the Clavien–Dindo classification, scaled from 0 (no complication) to V (most complications), to grade surgical complications. Other data included the ASA physical status score (ranging from I, indicating the lowest morbidity, to V, indicating the highest morbidity); diabetes history; metformin use; TNM stage of the primary tumor; histological grade of the tumor; R0 (margin-negative)/R1(margin-positive) resection; tumor size; intraoperative blood transfusion; intraoperative use of dexamethasone; postoperative chemotherapy; presence of postoperative recurrence; and presence of postoperative metastases. Because these variables have been shown or posited to affect patient outcomes, they were chosen as potential confounders. In our hospital, we follow the pancreatic ductal adenocarcinoma guidelines, and routinely combine tumor marker testing and computed tomography imaging every three to six months during the first two years after pancreatic cancer surgery [21]. In addition, we routinely use Gemzar (gemcitabine) as the postoperative chemotherapy for pancreatic cancer patients. And there was no difference in the two groups between the used chemotherapy regimen [22].

## Statistical methods

The primary end point was overall survival, which was compared between the propofol and desflurane groups. The survival time was defined as the interval between the date of surgery and the date of death or February 11, 2019, for those who were censored. All data are presented as mean ± standard deviation (SD) or number (percentage).

Mortality rates and patient characteristics were compared between the groups treated with the different anesthetics using Student's *t* test or the chi-square test. The survival according to the anesthetic technique was depicted visually in a Kaplan–Meier survival curve. The association between the anesthetic techniques (propofol or desflurane) and survival was analyzed by the Cox proportional-hazards model with and without adjustment for the abovementioned variables. Because significant interactions with the two anesthetic techniques (propofol or desflurane) were found, we also performed subgroup analyses for TNM stage, postoperative recurrence, and postoperative metastases formation.

Propensity score (PS) matching with IBM SPSS Statistics 22.0 was used to select for the most similar PSs for preoperative variables (with calipers set at 0.2 SD of the logit of the PS)

across each anesthesia: propofol or desflurane in a 1:1 ratio, to make sure the comparability between propofol and desflurane anesthesia before the surgery. Two-tailed *P*-values less than 0.05 were considered statistically significant.

## Results

The patient and treatment characteristics are shown in Table 1. Propofol anesthesia had longer time since the earliest included patient compared with desflurane anesthesia (8.2 ± 3.3 *vs* 6.2 ± 3.7 years; *P* = 0.001). Calendar periods were significantly different between the two groups (*P* < 0.001). Sex, age, CCI, diabetes history, metformin use, preoperative functional status, ASA score, TNM stage of the primary tumor, preoperative CA19-9 level, tumor size, histological grade of the tumor, R0/R1 resection, grade of surgical complications, need for intraoperative blood transfusion, intraoperative use of dexamethasone, and the use of postoperative chemotherapy were insignificantly different between the two anesthetic techniques (Table 1).

The overall mortality rate was significantly lower in the propofol anesthesia (60.0%) than in the desflurane anesthesia (82.0%) during follow-up (*P* = 0.006). Additionally, the cancer-specific mortality rate was significantly lower in the propofol anesthesia (57.0%) than in the desflurane anesthesia (78.0%) during follow-up (*P* = 0.014). A lower percentage of patients in the propofol anesthesia (43.0%) exhibited postoperative recurrence compared with the desflurane anesthesia (66.0%; *P* = 0.010). The presence of postoperative metastases did not differ between the two groups (Table 1). Kaplan–Meier survival curves for the two anesthetic techniques are shown in Fig 2A.

The overall mortality risk associated with the use of propofol and desflurane during pancreatic cancer surgery is reported in Table 2. Overall survival from the date of surgery grouped according to the anesthetic technique and other variables was compared individually in a univariable Cox model and subsequently in a multivariable Cox regression model. Other variables that significantly increased the mortality risk were higher CCI, higher TNM stage, higher preoperative CA19-9 level, no metformin use, and no intraoperative use of dexamethasone after the multivariable analysis (Table 2). Patients with propofol anesthesia was associated with improved overall survival compared to those with desflurane anesthesia (overall survival 40.0% versus 18.0%, respectively; the crude hazard ratio (HR) was 0.63 (95% confidence interval (CI), 0.42–0.93; *P* = 0.021). This finding did not change substantially in the multivariable analysis after adjustment for the time since the earliest included patient, CA19-9 level, CCI, ASA score, TNM stage, metformin use, postoperative chemotherapy, intraoperative use of dexamethasone, grade of surgical complications, and surgeons (HR, 0.53; 95% CI, 0.32–0.86; *P* = 0.010) (Table 2).

We used the PS from the logistic regression to adjust the baseline characteristics and the choice of therapy between the two anesthetic techniques due to the significant differences in baseline characteristics between the two anesthetic techniques. Fifty-eight pairs were formed after matching (Table 1). Patient characteristics and prognostic factors of pancreatic cancer were insignificantly different between the matched groups (except calendar period; Table 1). Kaplan–Meier survival curves for the two anesthetic techniques are shown in Fig 2B.

### Subgroup analyses for all-cause mortality, cancer-specific mortality, presence of postoperative metastasis, postoperative recurrence, TNM stage, and disease progression

In the all-cause mortality analyses, patients with propofol anesthesia showed better survival than those with desflurane; the crude HR was 0.63 (95% CI, 0.42–0.93; *P* = 0.021), and the PS-matched HR was 0.65 (95% CI, 0.42–0.99; *P* = 0.047) (Table 3).

**Table 1. Patients' and treatment characteristics and clinical outcomes for overall group and matched group after propensity scoring.**

| Variables | Overall Patients | | | Matched Patients | | |
|---|---|---|---|---|---|---|
| | Propofol | Desflurane | *P* value | Propofol | Desflurane | *P* value |
| | (n = 72) | (n = 68) | | (n = 58) | (n = 58) | |
| Time since the earliest included patient (years), Mean (SD) | 8.2 (3.3) | 6.2 (3.7) | 0.001 | 7.3 (3.1) | 6.7 (3.8) | 0.371 |
| Calendar period, n (%) | | | < 0.001 | | | 0.010 |
| 2005–2009 | 9 (13) | 28 (41) | | 9 (16) | 20 (35) | |
| 2010–2014 | 39 (54) | 25 (37) | | 39 (67) | 23 (40) | |
| 2015–2017 | 24 (33) | 15 (22) | | 10 (17) | 15 (26) | |
| Male sex, n (%) | 31 (43) | 33 (49) | 0.631 | 26 (45) | 27 (47) | 1.000 |
| Age (years), Mean (SD) | 62 (12) | 63 (12) | 0.754 | 62 (11) | 63 (12) | 0.913 |
| Charlson comorbidity index, Mean (SD) | 5.1 (1.3) | 5.0 (1.5) | 0.635 | 5.2 (1.3) | 5.0 (1.5) | 0.645 |
| Diabetes history, n (%) | 18 (25) | 24 (35) | 0.253 | 13 (22) | 19 (33) | 0.299 |
| Metformin use, n (%) | 8 (11) | 7 (10) | 1.000 | 5 (9) | 5 (9) | 1.000 |
| Functional status, n (%) | | | 0.704 | | | |
| < 4MET | 13 (18) | 15 (22) | | N/A | N/A | |
| ≥ 4MET | 59 (82) | 53 (78) | | N/A | N/A | |
| ASA, n (%) | | | 0.704 | | | 1.000 |
| II | 59 (82) | 53 (78) | | 46 (79) | 46 (79) | |
| III | 13 (18) | 15 (22) | | 12 (21) | 12 (21) | |
| TNM stage of primary tumor, n (%) | | | 0.552 | | | 0.759 |
| I | 10 (14) | 14 (21) | | 9 (16) | 12 (21) | |
| II | 48 (67) | 43 (63) | | 38 (65) | 35 (60) | |
| III | 14 (19) | 11 (16) | | 11 (19) | 11 (19) | |
| CA19-9, n (%) | | | 0.150 | | | 0.401 |
| ≤ 37 | 26 (36) | 16 (24) | | 18 (31) | 13 (22) | |
| > 37 | 46 (64) | 52 (77) | | 40 (69) | 45 (78) | |
| Tumor size, Mean (SD) | 3.5 (1.6) | 3.3 (1.2) | 0.363 | 3.3 (1.5) | 3.2 (1.2) | 0.625 |
| Tumor grade, n (%) | | | 0.594 | | | 0.646 |
| I | 14 (19) | 15 (22) | | 12 (21) | 14 (24) | |
| II | 40 (56) | 32 (47) | | 33 (57) | 28 (48) | |
| III | 18 (25) | 21 (31) | | 13 (22) | 16 (28) | |
| R0/R1 resection, margin-positive, n (%) | 14 (19) | 11 (16) | 0.777 | 11 (19) | 11 (19) | 1.000 |
| Grade of surgical complications, n (%) | | | 0.512 | | | 0.349 |
| 0 | 52 (72) | 43 (63) | | 43 (74) | 36 (62) | |
| I | 18 (25) | 22 (32) | | 13 (22) | 20 (35) | |
| II& III | 2 (3) | 3 (4) | | 2 (3) | 2 (3) | |
| Intraoperative blood transfusion, n (%) | 22 (31) | 24 (35) | 0.677 | 16 (28) | 18 (31) | 0.838 |
| Intraoperative dexamethasone use, n (%) | 55 (76) | 43 (63) | 0.130 | 42 (72) | 41 (71) | 1.000 |
| Postoperative chemotherapy, n (%) | 47 (65) | 45 (66) | 1.000 | 36 (62) | 39 (67) | 0.698 |
| Postoperative recurrence, n (%) | 31 (43) | 45 (66) | 0.010 | 27 (47) | 39 (67) | 0.039 |
| Postoperative metastasis, n (%) | 12 (17) | 9 (13) | 0.740 | 9 (16) | 9 (16) | 1.000 |
| All-cause mortality, n (%) | 43 (60) | 56 (82) | 0.006 | 37 (64) | 48 (83) | 0.036 |
| Cancer-specific mortality, n (%) | 41 (57) | 53 (78) | 0.014 | 35 (60) | 47 (81) | 0.025 |

Data shown as mean ± SD or n (%). Grade of surgical complications: Clavien-Dindo classification. MET = metabolic equivalents; ASA = American Society of Anesthesiologists; TNM = tumor–node–metastasis; N/A = not applicable.

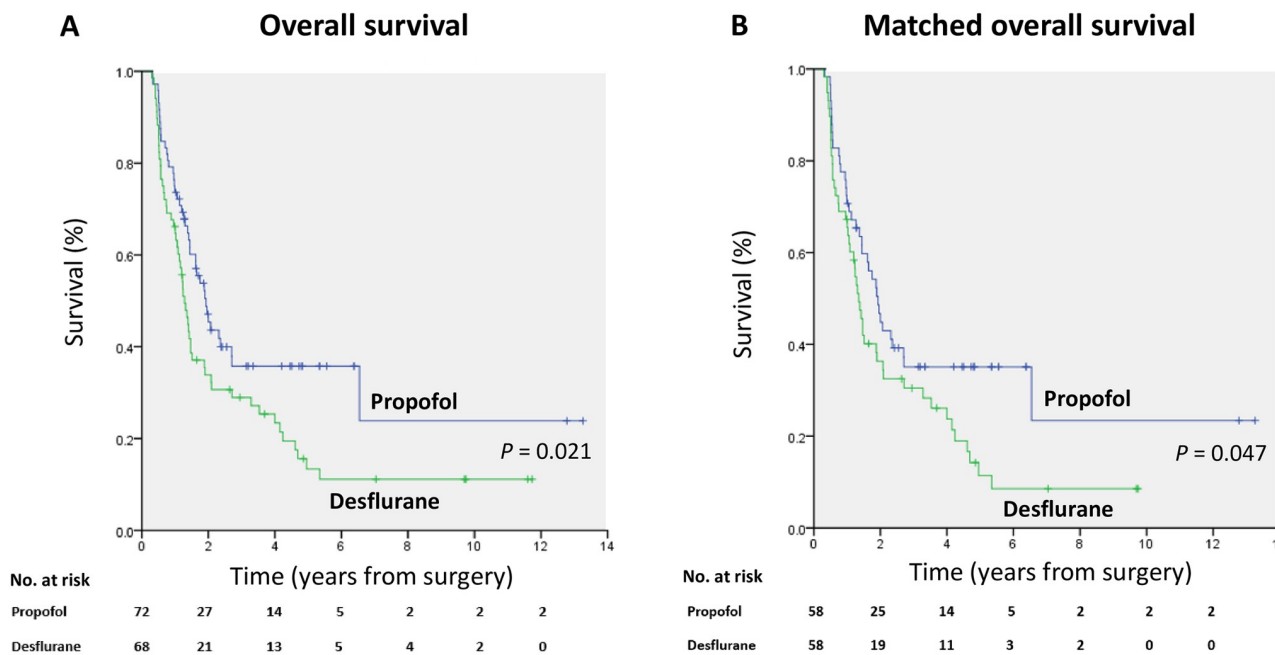

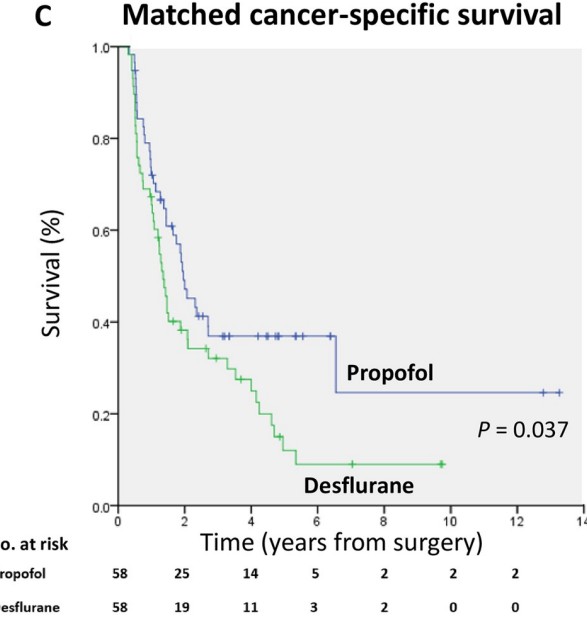

**Fig 2.** (A) Overall survival curves from the date of surgery by anesthesia type. (B) Overall survival curves from the date of surgery by anesthesia type after propensity score matching. (C) Cancer-specific survival curves from the date of surgery by anesthesia type after propensity score matching.

In the cancer-specific mortality analyses, patients with propofol anesthesia also showed better survival than those with desflurane anesthesia; the crude HR was 0.63 (95% CI, 0.42–0.95; *P* = 0.028), and the PS-matched HR was 0.63 (95% CI, 0.40–0.97; *P* = 0.037) (Table 3). Kaplan–Meier survival curves for the two anesthetic techniques are shown in Fig 2C.

**Table 2. Cox proportional hazards regression for mortality: Univariable and multivariable models for overall patients.**

| | Univariable | | Multivariable | |
|---|---|---|---|---|
| Variables | HR (95% CI) | *P* value | HR (95% CI) | *P* value |
| Anesthesia, Propofol (ref: Desflurane) | 0.63 (0.42–0.91) | 0.021 | 0.53 (0.32–0.86) | 0.010 |
| Time since the earliest op (years) | 0.94 (0.89–0.99) | 0.026 | 1.00 (0.93–1.08) | 0.983 |
| Female (ref: Male) | 1.07 (0.72–1.59) | 0.732 | | |
| Age (years) | 1.03 (1.02–1.05) | <0.001 | 0.98 (0.95–1.01) | 0.167 |
| Charlson comorbidity index | 1.41 (1.22–1.63) | <0.001 | 1.72 (1.30–2.29) | < 0.001 |
| Diabetes history (ref: No) | 1.23 (0.80–1.88) | 0.342 | | |
| Metformin use (ref: No) | 0.30 (0.12–0.74) | 0.009 | 0.21 (0.08–0.55) | 0.002 |
| Functional status, ≥4 METs (ref: <4 METs) | 0.48 (0.30–0.76) | 0.002 | | |
| ASA, III, (ref: II) | 2.09 (1.31–3.31) | 0.002 | 0.90 (0.48–1.69) | 0.733 |
| TNM stage of primary tumor (ref: I) | | | | |
| II | 2.80 (1.39–5.62) | 0.004 | 2.58 (1.20–5.55) | 0.016 |
| III | 7.33 (3.34–16.1) | <0.001 | 5.20 (2.14–12.6) | < 0.001 |
| CA19-9, >37 (ref: ≤ 37) | 3.89 (1.29–6.60) | <0.001 | 2.25 (1.24–4.09) | 0.008 |
| Tumor size | 0.95 (0.83–1.09) | 0.477 | | |
| Tumor grade (ref: I) | | | | |
| II | 2.29 (1.25–4.20) | 0.007 | | |
| III | 3.65 (1.90–7.01) | <0.001 | | |
| R0/R1, margin-positive (ref: margin-negative) | 3.13 (1.93–5.06) | <0.001 | | |
| Intraoperative blood transfusion (ref: no) | 1.19 (0.78–1.81) | 0.415 | | |
| Intraoperative dexamethasone use (ref: no) | 0.42 (0.28–0.63) | <0.001 | 0.42 (0.26–0.69) | 0.001 |
| Grade of surgical complications (ref: 0) | | | | |
| I | 2.20 (1.42–3.42) | <0.001 | 1.51 (0.81–2.81) | 0.191 |
| II& III | 4.52 (1.75–11.7) | 0.002 | 1.40 (0.51–3.82) | 0.513 |
| Postoperative chemotherapy (ref: no) | 1.58 (1.02–2.46) | 0.040 | 1.38 (0.83–2.32) | 0.219 |
| Postoperative recurrence (ref: no) | 4.24 (2.65–6.79) | <0.001 | | |
| Postoperative metastasis (ref: no) | 2.98 (1.78–4.98) | <0.001 | | |

Adjusted-HRs were adjusted by those variables were significant in the univariable analyses and surgeons (n = 8). Three variables were excluded from the multivariable due to they were highly correlated with other variables (functional status with ASA, Tumor grade and R0/R1 with TNM stage). MET = metabolic equivalents; ASA = American Society of Anesthesiologists; TNM = tumor–node–metastasis; N/A = not applicable.

There were no interaction between the anesthetic technique and postoperative metastases formation (*P* = 0.733), between the anesthetic techniques and postoperative recurrence (*P* = 0.324), and between the anesthetic techniques and TNM stage (*P* = 0.154), though the propofol anesthesia was associated with better outcomes in patients without postoperative metastasis (PS-matched HR, 0.59; 95% confidence interval, 0.37–0.96; *P* = 0.034) or with TNM II (PS-matched HR, 0.53; 95% confidence interval, 0.31–0.90; *P* = 0.018) (Table 3).

Patients with propofol anesthesia had less postoperative recurrence than those with desflurane; the crude HR was 0.53 (95% CI, 0.34–0.84; *P* = 0.007), and the PS-matched HR was 0.55 (95% CI, 0.34–0.90; *P* = 0.028). With regard to postoperative metastases formation, patients with propofol anesthesia showed no significant difference from those with desflurane anesthesia; the crude HR was 1.05 (95% CI, 0.44–2.50; *P* = 0.905), and the PS-matched HR was 0.83 (95% CI, 0.33–2.10; *P* = 0.695). Patients with propofol anesthesia had less postoperative recurrence and postoperative metastases formation than those with desflurane anesthesia; the crude HR was 0.62 (95% CI, 0.42–0.93; *P* = 0.019), and the PS-matched HR was 0.60 (95% CI, 0.39–0.93; *P* = 0.023) (Table 3).

**Table 3. Subgroup analyses for all-cause mortality, cancer-specific mortality, presence of postoperative metastasis, postoperative recurrence, TNM stage, and disease progression.**

| | Anesthesia | Crude-HR (95% CI) | *P* value | *P* value (interaction) | PS matched-HR (95% CI) | *P* value |
|---|---|---|---|---|---|---|
| **All-cause motality** | | | | | | |
| | Desflurane | 1.00 | | | 1.00 | |
| | Propofol | 0.63 (0.42–0.93) | 0.021 | | 0.65 (0.42–0.99) | 0.047 |
| **Cancer-specific mortality** | | | | | | |
| | Desflurane | 1.00 | | | 1.00 | |
| | Propofol | 0.63 (0.42–0.95) | 0.028 | | 0.63 (0.40–0.97) | 0.037 |
| **Postoperative metastasis** | | | | 0.733 | | |
| No | Desflurane | 1.00 | | | 1.00 | |
| | Propofol | 0.56 (0.36–0.88) | 0.012 | | 0.59 (0.37–0.96) | 0.034 |
| Yes | Desflurane | 1.00 | | | 1.00 | |
| | Propofol | 0.75 (0.29–1.97) | 0.565 | | 0.78 (0.29–2.12) | 0.625 |
| **Postoperative recurrence** | | | | 0.324 | | |
| No | Desflurane | 1.00 | | | 1.00 | |
| | Propofol | 0.57 (0.26–1.28) | 0.171 | | 0.60 (0.25–1.45) | 0.255 |
| Yes | Desflurane | 1.00 | | | 1.00 | |
| | Propofol | 1.01 (0.63–1.62) | 0.981 | | 1.07 (0.64–1.79) | 0.792 |
| **TNM stage** | | | | 0.154 | | |
| TNM: I | Desflurane | 1.00 | | | 1.00 | |
| | Propofol | 0.31 (0.06–1.50) | 0.145 | | 0.40 (0.08–2.06) | 0.271 |
| TNM: II | Desflurane | 1.00 | | | 1.00 | |
| | Propofol | 0.53 (0.32–0.86) | 0.010 | | 0.53 (0.31–0.90) | 0.018 |
| TNM: III | Desflurane | 1.00 | | | 1.00 | |
| | Propofol | 1.12 (0.46–2.72) | 0.806 | | 1.19 (0.47–2.97) | 0.717 |
| **Disease progression** | | | | | | |
| Postoperative recurrence | Desflurane | 1.00 | | | 1.00 | |
| | Propofol | 0.53 (0.34–0.84) | 0.007 | | 0.55 (0.34–0.90) | 0.028 |
| Postoperative metastasis | Desflurane | 1.00 | | | 1.00 | |
| | Propofol | 1.05 (0.44–2.50) | 0.905 | | 0.83 (0.33–2.10) | 0.695 |
| Postoperative recurrence + | Desflurane | 1.00 | | | 1.00 | |
| Postoperative metastasis | Propofol | 0.62 (0.42–0.93) | 0.019 | | 0.60 (0.39–0.93) | 0.023 |

HR = hazard ratio; PS = propensity score; TNM = tumor–node–metastasis.

In summary, patients with desflurane anesthesia had higher all-cause mortality, higher cancer-specific mortality, and poorer disease progression (such as postoperative recurrence, or postoperative recurrence and metastases) than those under propofol anesthesia.

## Discussion

The major finding in the present study is that propofol anesthesia in open pancreatic cancer surgery is associated with improved survival and lower risk of postoperative recurrence compared with desflurane. These findings are consistent with those of previous researches of propofol anesthesia that demonstrated better survival following surgery for gastrointestinal cancers, such as esophageal, intrahepatic cholangiocarcinoma, HCC, or colon cancer compared with VAs [16,17,23,24]. Groot et al. [25] reported that disease progression (either postoperative recurrence or postoperative metastases formation) of pancreatic cancer occurs in 80% of patients within 2 years after potentially curative resections. In this study, disease

progression in the matched propofol group was less than in the matched desflurane group after pancreatic cancer surgery (*P* = 0.023; Table 3).

Surgical resection is the gold standard therapy for solid, potentially resectable tumors. But surgery may suppress important host defenses and stimulate the development of metastases. After the pancreatic cancer surgery, the outcomes remain poor with a median survival of only 20 to 22 months from the date of diagnosis [2]. Postoperative metastases formation and cancer recurrence have impacts on patient prognosis and survival in pancreatic cancer; thus, studies on pancreatic cancer have focused on searching paths to ameliorate overall patient survival via reducing them [2]. The plausibility of tumor metastasis depends on the balance between the cancer metastatic potential and the host defense, of which natural killer cell function and cell-mediated immunity are important parts [26]. Data from studies of human cancer cell lines and animal showed that different anesthetic techniques or anesthetics might influence the immune system in different ways [4–9] and affect risks of cancer recurrence or metastasis or the cancer patient's survival [6,8–11].

In the literature, only one study had compared the effects of the use of desflurane versus propofol anesthesia on patient outcomes after pancreatic cancer surgery [15]. Soliz et al. [15] showed that propofol-based anesthesia was associated with no complication or a low-grade (grades 1 or 2) complication, but not recurrence or metastasis or mortality, compared with desflurane-based anesthesia in pancreatic cancer surgery. Here, we found a 35% lower death rate with propofol anesthesia compared with desflurane in pancreatic cancer surgery. We previously reported that propofol anesthesia was related to a lower incidence of postoperative recurrence and metastasis compared with desflurane anesthesia in colon cancer and HCC surgery [16,17]. By contrast, recent retrospective studies reported insignificant differences in overall survival between the use propofol and VAs [11,27,28]. There are very few researches of the effects of the anesthetic techniques in pancreatic cancer patients; further investigations are needed to illuminate the effects of the anesthetic techniques on pancreatic cancer recurrence and metastasis in pancreatic cancer surgery.

In this study, we found that a higher CCI score, a higher TNM stage, or a higher preoperative CA19-9 level were associated with poor survival after pancreatic cancer surgery, as has been observed previously [2,19,29]. We also found that intraoperative administration of dexamethasone was associated with improved survival in pancreatic cancer surgery, which is consistent with the previous studies [2,30]. The anti-inflammatory effects of dexamethasone may contribute to better survival [2,30], but further investigation is necessary. Finally, we found that metformin use was associated with improved survival in pancreatic cancer surgery, which is consistent with a recent meta-analysis [31].

Data from human pancreatic cancer cell lines support the influence of propofol on pancreatic cancer cell growth and survival via different pathways [32–35]. Chen et al. [32] reported that propofol suppressed vascular endothelial growth factor expression and the migration ability of pancreatic cancer cells via inhibiting the N-methyl-D-aspartate receptor. In addition, Wang et al. [33] revealed that propofol suppressed the proliferation and invasion of pancreatic cancer cells by upregulating microRNA-133a expression. Moreover, Liu et al. [34] found that propofol inhibited the growth and invasion of pancreatic cancer cells through the regulation of the miR-21/Slug signaling pathway. Recently, Yu et al. [35] reported that propofol inhibited pancreatic cancer proliferation and metastasis by upregulating miR-328 and downregulating ADAM8. These findings suggest that propofol may be a useful drug for treating pancreatic cancer, though further clinical studies are needed.

Previous research showed that isoflurane had deleterious effects on the upregulation of hypoxia-inducible factor (HIF) and stimulated angiogenesis in prostate and renal cancer cells [36,37]. Upregulation of HIF was associated with a poor prognosis in one clinical cancer study [38]. By contrast, propofol was reported to reduce HIF-1α expression in prostate cancer cells

[36]. HIF-1α was overexpressed in pancreatic cancer [39], and a knockdown of HIF-1α suppressed the metastasis of pancreatic cancer [40]. Taken together, these limited reports suggest that the administration of isoflurane [3] or sevoflurane [3,11,12] may stimulate tumor cell growth, whereas propofol has a beneficial effect by suppressing tumor cell growth. However, to our knowledge, the mechanism by which desflurane anesthesia influences the recurrence or metastasis of pancreatic cancer remains unknown.

There were some limitations in this study. First, it was retrospective and the 140 patients were not randomly allocated. However, we used all available patients from January 2005 to July 2018 from the medical center. Patient characteristics such as time since the earliest included patient differed significantly between the groups, and we conducted PS matching to address this issue. Second, different VAs may have different effects on pancreatic cancer. We analyzed only desflurane because it is the most frequently used VA in our hospital. Third, nonsteroidal anti-inflammatory drugs (NSAIDs) seem to be safe in pancreatic cancer surgery [41]. Because of the risk of life-threatening complications such as peptic ulceration [42], in our hospital, we do not routinely use NSAIDs during pancreatic cancer surgery. Fourth, information about opioid use, especially for postoperative pain control by anesthesiologists, was incomplete in the medical records in the study. However, the intraoperative use of opioids does not appear to affect long-term survival after pancreatic cancer surgery [2]. Fifth, the use of a perioperative epidural anesthesia and analgesia may improve survival [2,43,44]; however, we do not routinely perform regional anesthesia and analgesia in pancreatic cancer surgery in our hospital. Sixth, we analyzed only pancreatic adenocarcinomas because they are the most common histological type of pancreatic cancer [1]. Finally, calendar period was conducted and significantly different between the matched two groups (*P* = 0.010; Table 1). However, both "time since the earliest included patient" (*P* = 0.983; Table 2) and the calendar period (Supplementary Table 1) did not affect the outcome.

In conclusion, during open pancreatic cancer surgery, propofol anesthesia was associated with improved survival compared with desflurane anesthesia. Patients under desflurane anesthesia had more postoperative recurrence, but not postoperative metastasis formation.

## Supporting information

**S1 Table. "Time since the earliest included patient" was replaced with "Calendar period".**
(DOC)

**S2 Table. Anesthesiologists * type of anesthesia crosstabulation.**
(DOCX)

**S3 Table. Anesthesiologists (n = 15).**
(DOCX)

## Acknowledgments

The authors thank the Cancer Registry Group of Tri-Service General Hospital for the clinical data support.

## Author Contributions

**Conceptualization:** Hou-Chuan Lai, Zhi-Fu Wu.

**Data curation:** Meei-Shyuan Lee, Yin-Tzu Liu, Kuen-Tze Lin, Kuo-Chuan Hung, Jen-Yin Chen.

**Formal analysis:** Meei-Shyuan Lee, Kuen-Tze Lin, Kuo-Chuan Hung, Jen-Yin Chen.

**Investigation:** Hou-Chuan Lai, Yin-Tzu Liu, Zhi-Fu Wu.

**Methodology:** Hou-Chuan Lai, Meei-Shyuan Lee, Yin-Tzu Liu, Kuen-Tze Lin, Zhi-Fu Wu.

**Supervision:** Zhi-Fu Wu.

**Writing – original draft:** Hou-Chuan Lai.

**Writing – review & editing:** Zhi-Fu Wu.

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
