## [Decision Letter · Decision Letter 0]

29 Jan 2020

PONE-D-19-21803

Propofol-Based Intravenous Anesthesia Is Associated with Better Survival Than Desflurane Anesthesia in Pancreatic Cancer Surgery

PLOS ONE

Dear Dr. Wu,

Thank you for submitting your manuscript to PLOS ONE. After careful consideration, we feel that it has merit but does not fully meet PLOS ONE’s publication criteria as it currently stands. Therefore, we invite you to submit a revised version of the manuscript that addresses the points raised during the review process.

Please kindly respond to the questions raised by the reviewers. 

We would appreciate receiving your revised manuscript by Mar 14 2020 11:59PM. To enhance the reproducibility of your results, we recommend that if applicable you deposit your laboratory protocols in protocols.io, where a protocol can be assigned its own identifier (DOI) such that it can be cited independently in the future. For instructions see: http://journals.plos.org/plosone/s/submission-guidelines#loc-laboratory-protocols

We look forward to receiving your revised manuscript.

Kind regards,

Jason Chia-Hsun Hsieh, M.D. Ph.D

Academic Editor

PLOS ONE

Additional Editor Comments:

Please kindly respond to the questions raised by the reviewers.

2. We note you have included a table to which you do not refer in the text of your manuscript. Please ensure that you refer to Table 3 in your text; if accepted, production will need this reference to link the reader to the Table.

Reviewers' comments:

Reviewer's Responses to Questions

**Comments to the Author**

1. Is the manuscript technically sound, and do the data support the conclusions?

Reviewer #1: Yes

Reviewer #2: Partly

2. Has the statistical analysis been performed appropriately and rigorously? 

Reviewer #1: I Don't Know

Reviewer #2: Yes

3. Have the authors made all data underlying the findings in their manuscript fully available?

Reviewer #1: No

Reviewer #2: Yes

4. Is the manuscript presented in an intelligible fashion and written in standard English?

Reviewer #1: Yes

Reviewer #2: Yes

5. Review Comments to the Author

Reviewer #1: Comments to the author:

The authors report a retrospective cohort study of limited sample size addressing associations between chosen anesthetic medication and pancreatic cancer outcome. 140 patients who received open surgical resection for pancreatic cancer with either Propofol-based total intravenous anesthesia or desflurane-based balanced anesthesia were analyzed regarding overall mortality, cancer-specific mortality, local cancer recurrence and distant metastasis occurrence.

The main result of this retrospective analysis is, that Propofol-based anesthesia was associated with better overall, cancer specific survival and less local cancer recurrence but not distant metastasis occurrence after propensity score matching.

Major Comments:

• Introduction: You report that “Until now, very few studies have compared the effects of the use of desflurane versus propofol anesthesia on patient outcomes after pancreatic cancer surgery”. Please reference the respective work and better explain in the discussion what your study adds to the literature.

• You performed a retrospective analysis. Therefore, the nature of your study is exploratory and the findigs are descriptive and hypothesis generating. Throughout the entire manuscript please avoid any wording suggesting a causal relationship between type of anesthesia and outcome.

For example, Abstract: Use: “propofol anesthesia was associated with improved survival” instead of “propofol anesthesia improved survival”.

• The authors excluded patients who received sevoflurane anesthesia. However it would be of great interest whether the observed effects are specific for all volatile anesthetics or if they are restricted to desflurane. Sevoflurane Data should be reported.

Minor Comments:

• Introduction: Reference one is not sufficient to claim that pancreatic cancer has the worst prognosis of any cancer. Also, I am not sure if this is actually the case. How about malignant melanoma or glioblastoma?

• Were your 140 patients consecutive cases? If so, please add this information on page 11, (Participants and Data Sources) If not, how did you select individuals included in your analysis.

• P 11: “3–4 �g/mL in FiO2 of 100% oxygen at a flow rate of 300 mL/min”. FiO2 of 100% oxygen does not make sense for i.v. infusion of propofol.

• I recommend optimizing the consort diagram (figure 1): How many surgeries for pancreatic cancer were conducted in total in the study hospital between 2005 and 2019? How many patients were excluded due to use of other anesthetics (e.g. sevoflurane). This information would help to evaluate the expertise of the surgical center in sight of pancreatic cancer surgery.

• Since it is known that surgical resection margins have significant influence on pancreatic cancer outcome I recommend to doublecheck the two groups (propofol vs desflurane) for differences in surgical resection margins (R0 vs R1 resection). If available, data about R0 vs R1 resection margins should be added to the study group characteristics in Table 1.

• I recommend to separate baseline study characteristics from study outcome parameters (e.g. postoperative recurrence/metastasis) in Table 1 and show outcome parameters separately.

• Was there a specific protocol for postsurgical follow up? If a protocol exists please add information on whether CT scans were performed at pre-defined time intervals or if they were prompted on clinical suspicion.

• If available, please give information on the applied postoperative chemotherapies and whether there were differences in the two groups between the used chemotherapy regimen.

• What is the surgeon factor? (page 15)

• Data from table 3 are mentioned in abstract and manuscript but are not explicitly referred to in the text. Please add “(see Table 3 etc)” and make sure every table and diagram is named and referred to in the text.

• Avoid the expression “marginally significant”

• Due to the small sample size of approx. 10 surgeries per year I would like the authors to give more information on comparability of the study cohort with regard to surgical outcome and survival rates.

• page 15: please avoid the term “insignificant interaction”. Better: “We did not observe an interaction…”

• discussion: avoid presenting new data in the discussion (“disease progression in the matched propofol group(62,1%).

Reviewer #2: The authors performed a propensity matched analysis to investigate the influence of two anesthetics, namely desflurane and propofol. The topic is important and choosing the right anesthetic is modifiable factor in the anesthesia. The paper is concise and the statistical method and analysis seems feasible technique. I have the following concerns and advise.

1. I think diabetes and diabetic medication and other chronic medications (like metformin) might influence the survival. Do you have any data about it?

2. Regarding the Kaplan-Meier curves, the anesthetic techniques influence the long term outcome (after 3 years), could you give the number of the patients in risk in both groups below the diagram?Do you have an explanation for this phenomenon i.e. long term effect of the anesthetics?

3. Have you performed epidurals in the perioperative period?

4. Have you seen any random or fix effect of the different techniques duirng the years (have you changed the practice or was the choose of anesthetics left for individual discretion?

6. PLOS authors have the option to publish the peer review history of their article (what does this mean?). If published, this will include your full peer review and any attached files.

Reviewer #1: No

Reviewer #2: No

---

## [Author Response · Author response to Decision Letter 0]

14 Feb 2020

Please see "Response to Reviewers file", thank you.

---

## [Decision Letter · Decision Letter 1]

30 Mar 2020

PONE-D-19-21803R1

Propofol-Based Intravenous Anesthesia Is Associated with Better Survival Than Desflurane Anesthesia in Pancreatic Cancer Surgery

PLOS ONE

Dear Dr. Wu,

Thank you for submitting your manuscript to PLOS ONE. After careful consideration, we feel that it has merit but does not fully meet PLOS ONE’s publication criteria as it currently stands. Therefore, we invite you to submit a revised version of the manuscript that addresses the points raised during the review process.

ACADEMIC EDITOR: There are still some questions requiring to be addressed. Please complete the missing information and check them carefully. 

We would appreciate receiving your revised manuscript by May 14 2020 11:59PM. To enhance the reproducibility of your results, we recommend that if applicable you deposit your laboratory protocols in protocols.io, where a protocol can be assigned its own identifier (DOI) such that it can be cited independently in the future. For instructions see: http://journals.plos.org/plosone/s/submission-guidelines#loc-laboratory-protocols

We look forward to receiving your revised manuscript.

Kind regards,

Jason Chia-Hsun Hsieh, M.D. Ph.D

Academic Editor

PLOS ONE

Additional Editor Comments (if provided):

There still some questions requiring to be addressed. Please complete the missing information and check them carefully.

Reviewers' comments:

Reviewer's Responses to Questions

**Comments to the Author**

1. If the authors have adequately addressed your comments raised in a previous round of review and you feel that this manuscript is now acceptable for publication, you may indicate that here to bypass the “Comments to the Author” section, enter your conflict of interest statement in the “Confidential to Editor” section, and submit your "Accept" recommendation.

Reviewer #1: (No Response)

Reviewer #2: All comments have been addressed

2. Is the manuscript technically sound, and do the data support the conclusions?

Reviewer #1: Yes

Reviewer #2: Yes

3. Has the statistical analysis been performed appropriately and rigorously? 

Reviewer #1: I Don't Know

Reviewer #2: Yes

4. Have the authors made all data underlying the findings in their manuscript fully available?

Reviewer #1: Yes

Reviewer #2: Yes

5. Is the manuscript presented in an intelligible fashion and written in standard English?

Reviewer #1: Yes

Reviewer #2: Yes

6. Review Comments to the Author

Reviewer #1: Thank you for revising your manuscript and addressing our comments. We appreciate your reviced manuscript. However, there are a some comments that need further attention (see bullet Points below):

1) You added numbers at risk to diagram 2A-C as recommended by Reviewer2. Please double check Figure 2 A-C: If I am not mistaken the diagramm does not match with the underlaying values.

2) please check page 11:"3–4 �g /mL in FiO2 of 100% oxygen at a

flow rate of 300 mL/min”. FiO2 of 100% oxygen does

not make sense for i.v. infusion of Propofol

The wording is still confusing and we would recommend to separately describe Infusion concentrations of Propofol and inspiratory Oxygen fraction.

3) Your comment from response to reviewers: "In our hospital, we follow the pancreatic ductal

adenocarcinoma (PDAC) guidelines, and routinely

combine tumor marker testing and computed

tomography (CT) imaging every three to six months

during the first two years after pancreatic cancer

surgery. (Eur J Surg Oncol 2019;45:1770-1777)"

please also add this information in the text (e.g. methods"). Since metastasis occurance/tumor recurrence is monitored via CT/Tumor markers (time of diagnosis = time of diagnostic) it is important to know if all patients followed the same postoperative procedure. Otherwise it can not be distinguished whether metastasis occurance/tumor recurrence really occurs later in the propofol group or if the CT is just conducted later.

4) Your comment from response to reviewers: "In our hospital, we routinely use Gemzar

(gemcitabine) as the postoperative chemotherapy

for pancreatic cancer patients. And there was no

difference in the two groups between the used

chemotherapy regimen. (Am Fam Physician.

2014;89:626-32)"

Please add this information to the manuscript.

5) Page 16 line 1ff: Please double check "We did not observe an interaction between the anesthestic....(p=0.733) etc.: these values are not shown in table 3 as you refer to! Please add this o to table 3 or adjust the text.

Reviewer #2: My questions have been answered adequately , I think the paper is suitable for publication. The authors added new informations by the revision of the manuscript.

7. PLOS authors have the option to publish the peer review history of their article (what does this mean?). If published, this will include your full peer review and any attached files.

Reviewer #1: No

Reviewer #2: Yes: Andrea Szekely

---

## [Author Response · Author response to Decision Letter 1]

2 Apr 2020

Please refer to the attached file of "Response to Reviewers".

---

## [Decision Letter · Decision Letter 2]

11 May 2020

Propofol-Based Intravenous Anesthesia Is Associated with Better Survival Than Desflurane Anesthesia in Pancreatic Cancer Surgery

PONE-D-19-21803R2

Dear Dr. Wu,

We are pleased to inform you that your manuscript has been judged scientifically suitable for publication and will be formally accepted for publication once it complies with all outstanding technical requirements.

With kind regards,

Jason Chia-Hsun Hsieh, M.D. Ph.D

Academic Editor

PLOS ONE

Additional Editor Comments (optional):

After two rounds of revisions, the questions seemed to be answered adequately.

Reviewers' comments:

Reviewer's Responses to Questions

**Comments to the Author**

1. If the authors have adequately addressed your comments raised in a previous round of review and you feel that this manuscript is now acceptable for publication, you may indicate that here to bypass the “Comments to the Author” section, enter your conflict of interest statement in the “Confidential to Editor” section, and submit your "Accept" recommendation.

Reviewer #2: All comments have been addressed

2. Is the manuscript technically sound, and do the data support the conclusions?

Reviewer #2: Yes

3. Has the statistical analysis been performed appropriately and rigorously? 

Reviewer #2: Yes

4. Have the authors made all data underlying the findings in their manuscript fully available?

Reviewer #2: Yes

5. Is the manuscript presented in an intelligible fashion and written in standard English?

Reviewer #2: Yes

6. Review Comments to the Author

Reviewer #2: I have staisfied with the qulaity of the present manuscript. I have no more question. My questions and concerns have been adequately responded.

7. PLOS authors have the option to publish the peer review history of their article (what does this mean?). If published, this will include your full peer review and any attached files.

Reviewer #2: No

---

## [Editor Report · Acceptance letter]

13 May 2020

PONE-D-19-21803R2 

Propofol-Based Intravenous Anesthesia Is Associated with Better Survival Than Desflurane Anesthesia in Pancreatic Cancer Surgery 

Dear Dr. Wu:

I am pleased to inform you that your manuscript has been deemed suitable for publication in PLOS ONE. Congratulations! Your manuscript is now with our production department. 

With kind regards,

on behalf of

Dr. Jason Chia-Hsun Hsieh 

Academic Editor

PLOS ONE